# Interaction of Angiotensin II AT1 Receptors with Purinergic P2X Receptors in Regulating Renal Afferent Arterioles in Angiotensin II-Dependent Hypertension

**DOI:** 10.3390/ijms241411413

**Published:** 2023-07-13

**Authors:** Supaporn Kulthinee, Adis Tasanarong, Martha Franco, Luis Gabriel Navar

**Affiliations:** 1Department of Physiology, Hypertension and Renal Center of Excellence, Tulane University School of Medicine, New Orleans, LA 70112, USA; skulthinee1@tulane.edu; 2Chulabhorn International College of Medicine, Thammasat University, Klong Luang 12120, Thailand; tadis@tu.ac.th; 3Department of Cardio-Renal Physiopathology, Instituto Nacional de Cardiología “Ignacio Chávez”, Mexico City 14080, Mexico; marthafranco@lycos.com

**Keywords:** hypertension, P2X receptors, AT1 receptors, afferent arteriole, angiotensin II infusions

## Abstract

In angiotensin II (Ang II)-dependent hypertension, Ang II activates angiotensin II type 1 receptors (AT1R) on renal vascular smooth muscle cells, leading to renal vasoconstriction with eventual glomerular and tubular injury and interstitial inflammation. While afferent arteriolar vasoconstriction is initiated by the increased intrarenal levels of Ang II activating AT1R, the progressive increases in arterial pressure stimulate the paracrine secretion of adenosine triphosphate (ATP), leading to the purinergic P2X receptor (P2XR)-mediated constriction of afferent arterioles. Thus, the afferent arteriolar tone is maintained by two powerful systems eliciting the co-existing activation of P2XR and AT1R. This raises the conundrum of how the AT1R and P2XR can both be responsible for most of the increased renal afferent vascular resistance existing in angiotensin-dependent hypertension. Its resolution implies that AT1R and P2XR share common receptor or post receptor signaling mechanisms which converge to maintain renal vasoconstriction in Ang II-dependent hypertension. In this review, we briefly discuss (1) the regulation of renal afferent arterioles in Ang II-dependent hypertension, (2) the interaction of AT1R and P2XR activation in regulating renal afferent arterioles in a setting of hypertension, (3) mechanisms regulating ATP release and effect of angiotensin II on ATP release, and (4) the possible intracellular pathways involved in AT1R and P2XR interactions. Emerging evidence supports the hypothesis that P2X1R, P2X7R, and AT1R actions converge at receptor or post-receptor signaling pathways but that P2XR exerts a dominant influence abrogating the actions of AT1R on renal afferent arterioles in Ang II-dependent hypertension. This finding raises clinical implications for the design of therapeutic interventions that will prevent the impairment of kidney function and subsequent tissue injury.

## 1. Introduction

Angiotensin II (Ang II), the principal product of the renin angiotensin system, exerts a powerful role in the pathogenesis of hypertension and renal injury via the activation of angiotensin II type 1 receptors (AT1R), which are widely distributed in all regions of the kidneys [1,2,3]. Angiotensin II receptor blockers (ARBs) are recommended in the treatment of hypertension because they reduce blood pressure and the associated renal inflammation, fibrosis, and renal injury [4,5,6]. However, the pluripotent actions of Ang II involve interactions with many other vasoactive systems [7].

Extracellular nucleotides, particularly ATP, exert physiological and pathological actions via P2 purinergic receptors (P2XR and P2YR), which strongly influence renal vascular resistance, renal autoregulation, and tubular transport function [8,9,10,11,12]. However, the sustained overexpression and activation of purinergic receptors induces renal vasoconstriction and leads to the eventual development of glomerular and tubulointerstitial injury [13,14,15]. The participation of purinergic P2XR in the development and maintenance of hypertension-associated renal injury has progressively gained recognition [16]. In Ang II-dependent hypertension, P2X7R-mediated deleterious effects include the suppression of autoregulation and pressure natriuresis and the reduced oxygenation of the medulla [17]. Indeed, the P2X1R stimulation of afferent arteriolar vasoconstriction causes the reduction of glomerular blood flow and pressure [13]. Interestingly, in Ang II-dependent hypertension, the administration of P2X1R and P2X7R inhibitors restore the afferent arteriolar resistances back to normal values without reducing the blood pressure, which is maintained by the elevated systemic Ang II levels [18]. An important issue related to the influence of AT1 receptors under conditions of elevated intrarenal Ang II levels is the interaction between P2XR and AT1R in regulating afferent arteriolar resistance.

## 2. Regulation of Renal Afferent Arterioles in Ang II-Dependent Hypertension

The renal afferent arteriole is unique in terms of its responses to two major feedback mechanisms responsible for renal autoregulation, namely the myogenic mechanism and tubuloglomerular feedback mechanism (TGF). The segments near the glomerulus are regulated mainly by the TGF [19]. The diameters and luminal pressure in the segments closest to their origin have greater wall tension. Increases in renal perfusion pressure lead to rapid myogenic-mediated vessel wall contraction, which increases preglomerular resistance. Collectively, the myogenic and TGF mechanisms protect against glomerular barotrauma in acute as well as sustained hypertension [20]. Although renal autoregulation is impaired in hypertension [19,21], renal microvascular responses are able to protect the glomerular vasculature resetting and elevating the vascular tone of renal afferent arterioles via interactions between AT1R and P2XR [20].

In angiotensin II-dependent hypertension, Ang II activates AT1R throughout the body to increase systemic vascular resistance, leading to elevation of blood pressure. In particular, renal afferent arteriolar responsiveness to Ang II is enhanced in this stage [22], and the increase in renal afferent arteriolar resistance contributes to an initial adaptation of renal function. Renal AT1R activation is of cardinal importance in the development of Ang II-dependent hypertension and when AT1R are selectively deleted from the kidneys, the extrarenal AT1R are not sufficient to induce hypertension [23]. Although total kidney AT1R mRNA levels and receptor protein were not significantly increased after 2 weeks of Ang II infusion, they were sufficient to cause hypertension [24].

Nishiyama et al. [25] demonstrated that renal interstitial fluid Ang II levels were increased in Ang II-infused rats. Ang II levels in Ang II-infused rats were higher in renal cortical endosomes than in control rats via an AT1 receptor-mediated mechanism [26]. Li et al. [27] demonstrated that in AT1a receptor-deficient mice, AT1 receptor-mediated increases in Ang II in the kidney were prevented. These studies indicate that AT1Rs contribute to the augmentation of intrarenal Ang II and provide the basis for sustained maintenance of hypertension. Chronic Ang II infusion elicits sustained renal afferent arteriolar vasoconstriction, as supported by the augmentation of intrarenal Ang II [28], and the restoration of normal blood pressures through the vasodilator responses to AT1R blockade [29]. These findings confirm that the AT1Rs are not desensitized during chronic Ang II infusion for two weeks and play an important role in Ang II-dependent hypertension.

Under physiological conditions, the increases in renal perfusion pressure result in the augmentation of renal interstitial fluid ATP levels. A study using microdialysis to collect renal interstitial fluid showed that ATP was increased in response to elevation in renal perfusion pressure within the autoregulatory range in anesthetized dogs [30]. Furthermore, a study using biosensors for assessment of ATP in the renal cortex in response to changes in renal perfusion pressure in anesthetized Sprague Dawley rats showed that increases in renal perfusion pressure are associated with elevated interstitial concentrations of ATP [31]. The mechanisms involved are shear stress-dependent ATP release from endothelial cells of the renal microvasculature and ATP from macula densa cells during tubuloglomerular feedback responses, which collectively augment Ca^2+^ influx into vascular smooth muscle via the P2X receptor [20,32,33].

Under normotensive conditions, P2X1 receptors are expressed on the renal afferent arterioles but not in efferent arterioles [34]. In contrast, P2X7R activity is very low under physiological conditions [18]. Interstitial ATP activates P2X1R and elicits afferent arteriolar vasoconstriction in response to increases in RPP [35]. Moreover, renal autoregulation is impaired in P2X1R knockout mice [36]. In the juxtamedullary nephron preparation, superfusion with ATP at normal pressures resulted in afferent vasoconstriction, which was abolished completely by a P2X1R inhibitor; however, there was no significant effect of P2X7R inhibition [37]. Our results confirmed that the activation of P2X1R constricts renal afferent arterioles at normotensive pressures, whereas P2X7R expression is very low in normal rats. In contrast, both are overexpressed in Ang II-dependent hypertension [18,38], which indicates that both P2X1 and P2X7 receptors contribute to the renal adaptation to chronic Ang II infusion [39].

Collectively, two major mechanisms, Ang II via AT1R and ATP through P2X1R and P2X7R, (Figure 1) play critical roles in Ang II-dependent hypertension maintaining the elevated renal afferent arterial resistance. As described previously, treatment with AT1 receptor blockers reduced renal vascular resistance and decreased blood pressure back to normal levels indicating that AT1R activity is dominant and that the contribution of interstitial ATP associated with the reductions in arterial pressure is minimal. If the actions of the two systems were additive, the kidney’s vasculature would be under excessive vasoconstriction, which does not occur. One possible explanation that solves the conundrum is that P2XR and AT1R share intracellular signaling mechanisms to regulate renal vasoconstriction in Ang II-dependent hypertension. Two questions need to be answered to understand the mechanisms responsible for this regulation: (1) How do these two systems that regulate renal afferent arterioles interact via their respective receptors to share intracellular signaling mechanisms? and (2) Which system is dominant in regulating the renal afferent arterioles in sustained Ang II-dependent hypertension? Understanding how these mechanisms interact to reset myogenic tone and elevate renal vascular resistance could facilitate the design of therapeutic interventions that prevent the progression of renal injury in hypertension.

## 3. Functional Evidence of AT1R and P2XR Interactions in Regulating Renal Afferent Arterioles

Micropuncture studies of changes in glomerular dynamics in response to selective P2X1 and P2X7 receptor inhibition have demonstrated a dominance of P2XR on segmental vascular resistance in Ang II-induced hypertension. Interestingly, P2X1R and P2X7R blockers completely normalized afferent arteriolar resistance in Ang II-infused hypertensive rats, leading to an increased glomerular filtration rate and a single nephron plasma flow to normal or higher values, while the systemic hypertension was maintained [18]. These results indicate that purinergic P2X receptors effects were dominant in controlling renal vascular resistance in Ang II-induced hypertension. Furthermore, in rats receiving Ang II for 2 weeks in combination with Brilliant Blue G (BBG), a non-selective antagonist of P2X7 receptors, renal hemodynamics were associated with the reduced expression of P2X7R and decreased tubulointerstitial inflammation [38], indicating that P2X7R-mediated renal vasoconstriction is activated by the Ang II infusions. Menzies et al. [17] also found that after chronic Ang II treatment, Ang II induced focal P2X7R expression in the vascular smooth muscle of the preglomerular vessels. The specific P2X7R antagonist AZ11657312 increased renal medullary perfusion but only in Ang II-treated rats and not in control rats. Therefore, the activation of P2X7R induces microvascular vasoconstriction during sustained Ang II elevation.

Experiments utilizing the in vitro blood perfused juxtamedullary nephron preparation allowed the direct visualization of renal afferent arteriole dimensions in normal and hypertensive states. The kidneys were perfused with blood from normotensive and hypertensive rats. In kidneys from Ang II-dependent hypertensive rats, RPP was increased from 100 mmHg to 140 mmHg to simulate in vivo conditions [37]. When an AT1R blocker was super perfused, there was a small increase in afferent arteriolar diameter (AAD), indicating a minor chronic influence of AT1R maintaining afferent arteriolar diameter. In contrast, a blockade of P2X1R dilated AAD to near baseline values. Furthermore, the combination of P2X1R and P2X7R inhibitors restored AAD to control levels (Figure 2). The greater effect of the P2X7R inhibitor to significantly augment AAD in the kidneys from hypertensive rats was associated with a greater expression of P2X7R in afferent arterioles of hypertensive rats [18,38].

In response to superfusion with Ang II, kidneys from normotensive rats exhibited afferent arteriole vasoconstriction. An increase in RPP to 140 mmHg caused a further decrease in AAD. P2X1R blockade partially restored AAD to a value near baseline RPP. With the addition of the AT1R inhibitor, AAD returned to baseline values. These results indicate a predominant P2X1R effect on the autoregulatory component along with a maintained influence of AT1R in the regulation of AAD during elevation in arterial pressure and acute treatment with Ang II (Figure 2).

Taken together, these results provide functional evidence that renal P2X1R and P2X7R exert dominant roles in regulating afferent arterioles in kidneys from Ang II-induced hypertensive rats, which mitigate the AT1R influence on afferent arteriolar vasoconstriction induced by Ang II. The cellular mechanisms mediating the interactions between P2XR and AT1R remain unclear; however, several possible pathways can be explored.

## 4. Mechanisms Regulating ATP Release and Effect of Angiotensin II on ATP Release

Renal interstitial ATP levels are regulated by several mechanisms including mechano-transduction-mediated ATP release from the microvasculatrure and tubuloglomerular feedback-mediated ATP release from macula densa cells [20]. Increased renal perfusion pressure induces endothelial sheer stress, which triggers ATP release. Vascular smooth muscle cells may also release ATP. The mechanism by which the shear stress evokes ATP release is demonstrated in an elegant study by Yamamoto et al. [32]. These authors used real-time imaging fluorescence resonance energy transfer-based ATP biosensors. The exposure of endothelial cells to shear stress activates mitochondrial ATP generation, which leads to ATP release in the caveolae, triggering purinergic Ca^2+^ signaling. As shown in Ang II-infused hypertensive rats, in which systemic hypertension induced an increase in renal perfusion pressure causing increased sheer stress, which triggers ATP release. In addition, ATP released from macula densa cells acts directly through the stimulation of P2 receptors, which augments TGF-mediated vasoconstrictor responses, and the further metabolism of ATP to adenosine activates A1 receptors on afferent arterioles, contributing to the vasoconstriction [20]. There is a direct relationship between arterial pressure and renal interstitial fluid ATP concentrations, thus mediating the autoregulation of afferent arteriolar resistance [30].

The increases in renal interstitial fluid Ang II concentrations seen acutely also occur during chronically elevated arterial pressure, as occurs during chronic Ang II infusions [39]. Furthermore, Ang II has direct effects on ATP release. Palygin et al. [40] studied freshly isolated rat kidneys, infused with Ang II (1 μM), and under constant laminar flow, and the real-time measurements of endogenous substances were performed using an enzymatic biosensor technique. The results demonstrated the perfusion of the kidney with the Ang II-induced release of ATP and H_2_O_2_. Moreover, AT1 receptor antagonism with losartan (10 μM) inhibited the release of both ATP and H_2_O_2_ in response to Ang II infusion. These experiments clearly showed the Ang II effect on ATP release. In a further study [41], the authors found that Ang II induced the rapid release of both ATP and H_2_O_2_, which was greater in Dahl salt-sensitive rats compared to Sprague Dawley rats fed a low-salt diet. These results suggest that ATP and H_2_O_2_ are also critical in the development of salt-sensitive hypertension. Ang II induces NADPH oxidase activity, which results in superoxide production and a rapid increase in interstitial H_2_O_2_ concentration along with the elevation of intracellular Ca^2+^ concentration, which in turn stimulates ATP release to extracellular fluid.

In cultured smooth muscle cells from the guineapig taenia coli, treatment with 10 µM Ang II increased ATP release about four-fold above the baseline level. This was also demonstrated in the whole segments, as the administration of Ang II (0.3–3 µM) elicited the release of ATP from the segments of guineapig taenia coli. The peak of ATP release was observed 3 min after beginning of the administration of the Ang II. The maximum ATP release was approximately 3.6-fold more than the basal concentration via the administration of Ang II at 3 µM. The ATP concentration evoked by Ang II was abolished by losartan in both cultured smooth muscle cells and segments. These findings indicate that Ang II releases ATP from the smooth muscles through the activation of AT1R [42]. Intracellular signal transduction pathways involved in Ang II evoked ATP liberation were investigated in cultured smooth muscle cells. The release of ATP by Ang II was suppressed by an AT1 receptor antagonist but not by an AT2 receptor antagonist. Interestingly, the evoked release of ATP was almost completely inhibited in the presence of a PLC inhibitor, and a Ca^2+^-ATPase inhibitor. Furthermore, Ang II enhanced IP3 accumulation. These results suggest that intracellular Ca^2+^ signals activated via the stimulation of IP3 receptor are involved in the production of ATP evoked by Ang II [43].

Connexin43 (Cx43) and Pannexin 1 (Panx1) channels have been extensively studied as crucial pathways for the production of ATP [44]. In cell lines derived from mesangial cells, Ang II stimulation increased a substrate of RhoA/ROCK and Cx43; this response was followed by an increase in Panx1 and P2X7R. It seems that the activation of P2X7R leads to the Panx1 channel opening [45]. Thus, the opening of connexins and/or pannexins mediated by Ang II induces the ATP release to the extracellular fluid, with the subsequent activation of P2 receptors. This process induced Ca^2+^ overload, leading to vasoconstriction.

Collectively, the increases in arterial pressure exacerbate the direct effects of Ang II on ATP release. Thus, in Ang II-dependent hypertension, ATP is released by the increased sheer stress and macula densa-mediated mechanisms, causing the stimulation of P2XR on renal afferent arterioles, then Ang II activates AT1R and may share intracellular signaling pathways, regulating the renal afferent arteriole tone. In addition, the direct effect of Ang II on ATP release exerts crucial roles in physiopathological conditions such that ATP via P2XR is responsible for renal afferent arteriole dysfunction in hypertensive states.

## 5. Intracellular Pathways Mediating Interactions between AT1 and P2X Receptors

AT1R is a G protein (heterotrimeric guanine nucleotide-binding protein)-coupled receptor (GPCR), which binds to the heterotrimeric Gα_q_/11 protein to induce the hydrolysis of phosphatidylinositol 4,5-bisphosphate (PIP_2_) and the production of inositol 1,4,5-trisphosphate (IP3) and diacylglycerol through the activation of phospholipase C-β (PLC). IP3 induces intracellular Ca^2+^ mobilization from the endoplasmic reticulum to the cytosol, which increases Ca^2+^ entry leading to the activation of myosin light chain kinase to cause vascular smooth muscle contraction [46]. In contrast, P2XRs are ligand-gated ion channels that augment Ca^2+^ entry upon ATP binding; thus, the activation of P2XR causes the influx of Ca^2+^ from the extracellular compartment [47], which leads to further increases in Ca^2+^ release from intracellular stores. The increase in cytosolic Ca^2+^ concentration induces the opening of VDCC-dependent Ca^2+^ channels that contribute to further increases in the intracellular Ca^2+^ concentrations [48]. The P2XR intracellular domain contains a dual cluster of basic amino acids, which enables PIP_2_ binding [49] and potentiates the activities of P2X1R and P2X7R. This proposed pathway suggests that AT1R shares signaling pathways with P2X1R and P2X7R involving PIP_2_. PIP_2_ plays a role as a cofactor of both P2XR and AT1R receptor activation, which can influence channel opening activity. In addition, the production of PIP2 appears to regulate via β-arrestin, since phosphatidylinositol 4-phosphate 5-kinase (PIP5K) is required to bind with β-arrestin2 to produce PIP2 [50]. Purinergic receptor activation may also lead to the activation of β arrestin, as it was reported that treatment with ATP increased the binding of β-arrestin-2 to the submembranous domains of P2X7R [51], which could then be a dominant regulator of PIP2 activation.

Alternatively, it should be noted that β-arrestins act as GPCR adaptor proteins to terminate G protein signaling for receptor internalization and signal desensitization [52]. Interestingly, AT1R can also signal via G protein-independent mechanisms through β-arrestin. TRV120027, a biased agonist which selectively engages β-arrestin downstream of the AT1R, prevented the adverse effects of the G protein signaling pathway during hypertension [53]. β-arrestin2 knockout mice exhibited an exacerbated pressor response when placed on a high salt diet plus DOCA treatment [54]. Therefore, the activation of the Ang II-AT1R- β-arrestin pathway would serve as a physiological influence to counterbalance the responses to chronic angiotensin II infusions.

In spontaneously hypertensive rats (SHR), the preglomerular microcirculation is hypersensitive to angiotensin Ang II; the underlying mechanism is likely due to the interaction between G protein subunits αq and βγ subunits to activate PLC. The key interaction is the receptor for activated C kinase 1 (RACK1; a scaffolding protein) between Gβγ, Gαq, and PLC. In RACK1 knockdown in SHR, the preglomerular vascular smooth muscle cells had attenuated contractions to Ang II [55]. However, no direct P2XR–RACK1 interaction was reported; in this regard, the activation of PLC by P2X7R has been suggested as a mechanism [56]. Additionally, a modulation of P2X7R through the depletion of PIP2 has been reported [56].

Another potential interaction is with the RhoA/Rho kinase pathway, RhoA activates Rho kinase, which is an inhibitory myosin light chain phosphatase. This sequence stimulates vascular smooth muscle contraction [57]. The AT1R signaling pathway is not limited to Gα_q_/11 protein but also activates RhoA/Rho kinases pathway via G_12/13_. In Ang II-infused rats, Rho kinase inhibition led to greater vasodilation compared with control rats [58,59]. These data suggest that the AT1R-mediated activation of the Rho kinase pathway contributes to an increased afferent arteriolar resistance. In addition, the P2X1R agonist, α, β-methylene ATP reduced AAD, but this response was eliminated during Rho kinase inhibition [60]. These data suggest that the stimulation of the Rho kinase pathway is associated with P2XR activation leading to vasoconstriction.

It is currently recognized that receptor–receptor interactions (RRI) involve GPCRs [61]. AT1R activation forms complexes with other receptors [62,63,64,65,66,67,68], suggesting cross-regulation between AT1R and other signaling pathways. AT1Rs form receptor complexes with the B_2_ bradykinin receptor in renal mesangial cells [65] and with P2Y6 receptors in mouse smooth muscle cells [68]. RRI involving receptors from other families may also participate—P2X7R has a long COOH-terminal tail, a structural factor that facilitates physical interaction with other receptors [47]. When the receptors undergo dimerization, there are three possible configurations that the complex can assume: (A) the modulation of binding and the (B) modulation of active site and of the signaling cascade or (C) novel allosteric sites. Changes in the signaling cascade as the patten B associated with GPCR complex formation have been reported [61]. This aspect would be of particular importance in the AT1R–P2XR interaction that can regulate signaling pathways, such as distinct G protein classes or β-arrestins. However, the RRI interactions between AT1R and P2XR as applied to the renal microcirculation have not been studied. These issues warrant further research.

Taken together, these diverse factors may strongly influence the interactions between AT1R and P2XR in ANG II-dependent hypertension. Figure 3 schematically summarizes some of the potential signaling consequences occurring when the receptors interact.

## 6. The Involvement of NLRP3 Inflammasome in Angiotensin II-Dependent Hypertension

The NLRP3 inflammasome is a cytosolic protein complex that contains NLRP3 (nucleotide-binding oligomerization domain, leucine-rich repeat and pyrin domain containing 3) proteins, ASC (apoptosis-associated speck-like protein containing a caspase recruitment domain), and caspase-1 [69]. The activation mechanism of the NLRP3 inflammasome is complex because it is activated by a variety of danger signals, in particular extracellular ATP [70]. An increase in kidney NLRP3 mRNA was demonstrated in Ang II-induced hypertensive rats and immunosuppression via treatment with mycophenolate mofetil, attenuated the NLRP3 increase [71]. In the renal medulla, NLRP3 and ASC were highly expressed. Moreover, the medullary infusion of NLRP3 inflammasome activator, monosodium urate (MSU), resulted in a significant decrease in medullary blood flow without changes in mean arterial blood pressure. In addition, in both NLRP3 and ASC gene knockout mice, MSU failed to produce any reduction in renal medullary blood flow [72]. Thus, the reduction in renal medullary blood flow induced via the infusion of MSU is associated with NLRP3 inflammasome activation. These findings suggest that the NLRP3 inflammasome may be a critical regulatory as well as inflammatory mechanism, which may play an important role in the control of medullary blood flow and the development of renal injury.

Extracellular ATP activating P2X7R is recognized as a potent stimulus for the activation of the NLRP3 inflammasome [73]. P2X7R stimulation induces a fall of the intracellular K^+^ concentration due to its ability to activate an intrinsic large conductance plasma membrane pore [74]. P2X7R may interact with the NLRP3 inflammasome scaffold protein and colocalize to discrete cytoplasmic subplasmalemmal regions. P2X7R activation drives the recruitment of NLRP3 and enhances P2X7R/NLRP3 colocalization at sites where cytoplasmic Ca^2+^ increases and K^+^ falls. These observations partially clarify the mechanism of P2XR–NLRP3 inflammasome activation [75]. When activated, these local inflammatory mediators exert a critical influence on renal function. Thus, in Ang II-induced hypertension, Ang II downstream mechanisms promote inflammatory cell infiltration, proinflammatory cytokine release and the activation of the NLRP3 inflammasome, which further induces inflammatory reactions leading to the development and progression of renal tissue injury [13,18,76,77,78]. Chronic Ang II infusion results in hemodynamic alterations and renal inflammation associated with an overexpression of P2X7R. In this setting, P2X7R receptor blockade improves glomerular hemodynamics, reduces immune cell infiltration, proinflammatory cytokines and NLRP3 expression [38]. These findings raise the possibility that P2X7R partially controls the medullary blood flow by activating the NLRP3 inflammasome in ANG II-dependent hypertension. However, the functional significance of the inflammasome in renal afferent arteriole regulation remains poorly understood, and the mechanisms mediating the activation and regulation of NLRP3 inflammasome in Ang II-dependent hypertension need to be defined.

## 7. Conclusions

In this brief review, we discuss the interesting conundrum related to the interactions between the AT1R and purinergic receptors (P2X1R and P2X7R) in the regulation of renal afferent arterioles in Ang II-dependent hypertension with sustained vasoconstriction. Clearly, the systemic as well as the renal activation of AT1R is the dominant hypertensinogenic stimulus responsible for the hypertension and renal vasoconstriction, including sustained afferent arteriolar vasoconstriction. During this stage, AT1R inhibition dilates the afferent arterioles, demonstrating the sustained actions of AT1R. Nevertheless, the inhibition of P2X1R and P2X7R also dilates the afferent arterioles and abrogates the influence of AT1R, demonstrating the dominant role that P2XRs exert on the renal afferent arterioles in this condition. The evidence described above support the P2XR relevance in control of renal afferent arterioles in a rich Ang II milieu. This dual control of afferent arteriolar resistance in hypertension indicates that there is receptor or post-receptor convergence of signaling pathways leading to the contractile response. Although the specific underlying pathways have not been delineated, we discuss various possible signaling pathways including PIP2, RhoA/Rho kinase, and receptor–receptor interactions responsible for the convergence. Since the renal afferent arterioles primarily regulate renal vascular resistance and glomerular capillary pressure, the alterations of these microvessels in hypertension, while initially protective, may lead to renal injury, as observed in sustained Ang II-dependent hypertension. The interaction between AT1R and P2XR mechanisms that underlie the transition in the control of renal vascular resistance to P2XR may contribute to the activation of inflammatory factors, leading to renal injury and the development and progression of chronic kidney disease.

## Figures and Tables

**Figure 1 ijms-24-11413-f001:**
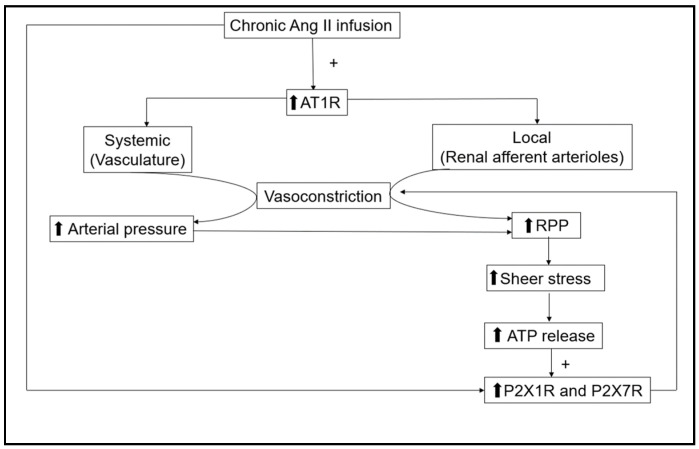
Actions of chronic angiotensin II (Ang II) infusion on systemic and kidney vasculature. Renal afferent arterioles are regulated via the direct effect of Ang II through AT1R and, secondarily, by P2X1 and P2X7 receptors due to a rise in ATP interstitial fluid concentrations caused by the increases in renal perfusion pressure (RPP). The increasing arrows represent the elevation of those factors.

**Figure 2 ijms-24-11413-f002:**
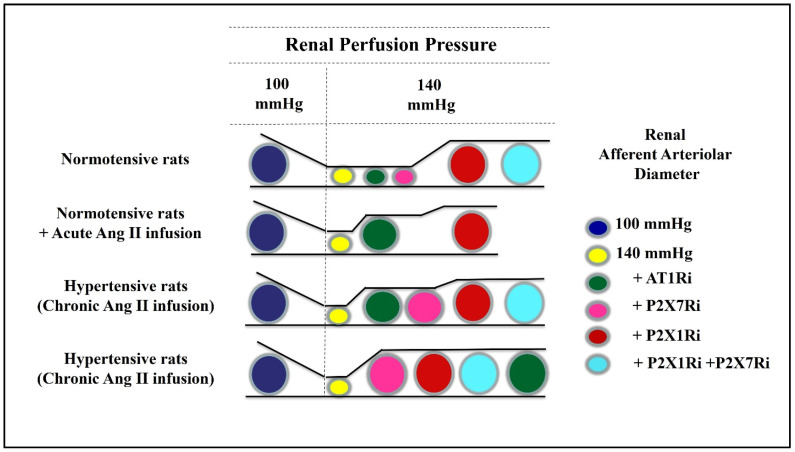
A schematic model illustrating the renal afferent arteriole diameter responses to renal perfusion pressure at 100 mmHg and 140 mmHg and the consequent impact of AT1 receptor inhibitor (AT1Ri), P2X7 receptor inhibitor (P2X7Ri), P2X1 receptor inhibitor (P2X7Ri), and P2X1 receptor inhibitor plus P2X7 receptor inhibitor (P2X1Ri + P2X7Ri).

**Figure 3 ijms-24-11413-f003:**
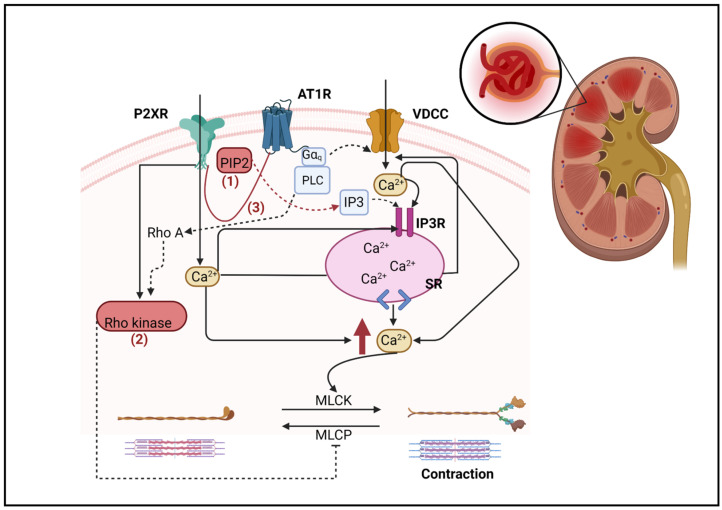
Proposed mechanisms for AT1R and P2XRs interaction. (1) AT1R may share important signaling pathways with P2X1R and P2X7R involving PIP_2_, (2) Rho kinase may contribute to AT1R and P2XR signaling interactions, and (3) physical interaction between AT1R and P2XR to mediate downstream signaling pathways. PIP2—phosphatidylinositol 4,5-bisphosphate; VDCC—voltage-dependent calcium channel; PLC—phospholipase C; IP3-inositol-1,4,5-triphosphate; IP3R—inositol-1,4,5-triphosphate receptor; SR—smooth endoplasmic reticulum. MLCP—myosin light chain phosphatase; MLCK—myosin light chain kinase. Dashed lines point out possible interactions; see text for more detail and references. Solid lined arrows represent signaling pathways and dashed lined arrows point out possible interactions. Created with BioRender.com.

## Data Availability

Not applicable.

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
