# Peer review of "Interaction of Angiotensin II AT1 Receptors with Purinergic P2X Receptors in Regulating Renal Afferent Arterioles in Angiotensin II-Dependent Hypertension"

_ijms, 2023, doi:10.3390/ijms241411413_

Round 1

Reviewer 1 Report

my opinion for the article after reading it is very positive.    It deals with the interactions between AT2R  and purinergic  receptors revealing important mechanisms that contribute to the activation of inflammatory factors.    Article sheds light to the dominant roles of purinergic receptors In the regulation of Ang II dependent hypertension providing further insights.   Article is well written and is a significant contribution to the field.   I recommend publication as it is.

Reviewer 2 Report

Kulthinee and co-workers focused on the interaction of angiotensin II (Ang II) type 1 receptor (AT1) and purinergic type 2X receptors in afferent arterioles in Ang II dependent hypertension. This is an interesting and important topic, since pathogenesis and pathophysiology of hypertension are still not fully understood. The kidney is a key organ in the development and maintenance of hypertension. Renal microvasculature may play an important role in this context. Afferent arterioles contribute to 50% to the total renal resistance and are of particular interest. An excessive activity of the renin angiotensin system (RAS) is a cause of hypertension in a substantial proportion of patients. The AT1 receptors mediate several important actions of Ang II such as vasoconstriction, inflammation, and fibrosis. Several studies showed an interaction of RAS and AT1, respectively, with the sympathetic and purinergic system in afferent arterioles.

The authors submit a concise review, which deals with important topics.

There are two points, which I would like to make: The relative low number of articles/studies in the last 5 years in this field and the speculative part regarding the molecular interactions of AT1 and P2X.